# Modulatory L-Alliin Effect on Acute Inflammatory Cytokines in Diet-Induced Obesity Mice

**DOI:** 10.3390/metabo14110580

**Published:** 2024-10-27

**Authors:** Daniel Ulises Torres-Reyes, Marina Alma Sánchez-Sánchez, Carmen de la Rocha, Argelia Esperanza Rojas-Mayorquín, Rocío Ivette López-Roa, Daniel Ortuño-Sahagún, Lucrecia Carrera-Quintanar

**Affiliations:** 1Programa de Doctorado en Ciencias de la Nutrición Traslacional (DCNT), Departamento de Reproducción Humana Crecimiento y Desarrollo Infantil, Centro Universitario de Ciencias de la Salud (CUCS), Universidad de Guadalajara (UdeG), Guadalajara 44340, Mexico; daniel.treyes@alumnos.udg.mx (D.U.T.-R.); carmen.martindc@academicos.udg.mx (C.d.l.R.); 2Departamento de Clínicas Médicas, Centro Universitario de Ciencias de la Salud (CUCS), Universidad de Guadalajara (UdeG), Guadalajara 44340, Mexico; marina.sanchez@academicos.udg.mx; 3Departamento de Biología Molecular y Genómica, Centro Universitario de Ciencias de La Salud (CUCS), Universidad de Guadalajara (UdeG), Guadalajara 44340, Mexico; 4Departamento Materno-Infantil, Centro Universitario de Tlajomulco (CUTlajo), Universidad de Guadalajara (UdeG), Tlajomulco 45641, Mexico; argelia.rojas@academicos.udg.mx; 5Laboratorio de Investigación y Desarrollo Farmacéutico (LIDF), Departamento de Farmacobiología (CUCEI), Universidad de Guadalajara (UdeG), Tlaquepaque 44430, Mexico; rocio.lopez@academicos.udg.mx; 6Laboratorio de Neuroinmunobiología Molecular, Instituto de Investigación en Ciencias Biomédicas (IICB), Centro Universitario de Ciencias de la Salud (CUCS), Universidad de Guadalajara (UdeG), Guadalajara 44340, Mexico

**Keywords:** S-allyl-cysteine sulfoxide, L-Alliin, obesity, inflammatory diseases, anti-inflammatory, nutraceuticals, functional foods, garlic

## Abstract

**Background/Objectives:** The inflammatory response has evolved as a protective mechanism against pathogens and tissue damage. However, chronic inflammation can occur, potentially leading to severe disease. Low-grade chronic inflammation is associated with obesity, and the Th1 cytokine profile plays an important role in this proinflammatory environment. Diet-induced obesity (DIO) can lead to persistent dysbiosis and maintain high concentrations of circulating lipopolysaccharides (LPSs) over prolonged periods of time, resulting in metabolic endotoxemia. In this context, the study of natural immunomodulators has recently increased. **Objective:** The aim of this study is improve scientific evidence for the immunomodulatory role of L-Alliin in obesity and inflammation. **Methods:** In the present work, we describe the effect of L-Alliin on serum levels of cytokines in DIO mice after an acute inflammatory challenge. L-Alliin is the main organosulfurized molecule of garlic that has been studied for its numerous beneficial physiological effects in health and disease and is beginning to be considered a nutraceutical. Two situations are simulated in this experimental model, health and chronic, low-grade inflammation that occurs in obesity, both of which are confronted with an acute, inflammation-inducing challenge. **Results:** Based on our findings, L-Alliin seems to somehow stimulate the cellular chemotaxis by eliciting the release of key molecules, including IL-2, IFN-γ, TNF-α, MCP-1, IL-6, IL-9, and G-CSF. However, the molecular mechanism involved remains unknown. This, in turn, mitigates the risk of severe inflammatory symptoms by preventing the release of IL-1β and its downstream molecules such as IL-1α, GM-CSF, and RANTES. **Conclusions:** Taken together, these results indicate that L-Alliin can boost immunity in healthy organisms and act as an immunomodulator in low-grade inflammation.

## 1. Introduction

Since 1990, global adult obesity has more than doubled, with 43.5% of individuals aged 18 and over classified as overweight [1] and 16% as obese [2]. In the same fashion, since 1990, the worldwide prevalence of deaths attributed to cardiovascular diseases has increased by 61% (from 12,345,727 to 19,906,615) [3]. Therefore, these diseases and their complications have become an enormous burden on health services and a worldwide public health problem. 

The inflammatory response has evolved as a protective mechanism against pathogens and tissue damage. However, if the natural self-limiting inflammatory process and its resolution mechanisms do not function properly, chronic inflammation can occur, potentially leading to severe disease [4]. At a systemic level, metaflammation refers to the low-grade chronic inflammation associated with obesity, which has systemic effects and affects multiple organs [5]. Therefore, obesity has attracted much attention as a global health problem, especially because of its associated comorbidities such as non-alcoholic fatty liver disease, type 2 diabetes, cardiovascular disease, cancer [6], and neurodegenerative diseases such as Alzheimer’s disease [7] and cognitive impairment [8]. 

The immune profile in obesity is characterized by the presence of M1, Th1 cells, and cytotoxic CD8 T cells. This is also promoted by the concentration of adipokines such as adiponectin and resistin, which inhibit Th2 cells and promote the Th1 profile [9]. The Th1 cytokine profile plays an important role in the proinflammatory environment in obesity. IFN-γ stimulates M1 macrophage polarization and is associated with adipose tissue inflammation and metabolic dysfunction [10]. Th2 cells, on the other hand, typically secrete IL -3, IL -4, IL -5, and IL -13 and are associated with a homeostatic context in adipose tissue, and the presence of Th2 cells is negatively correlated with systemic inflammation and insulin resistance in mice and humans [10]. 

Diet-induced obesity can alter the gut microbiota and intestinal permeability, leading to persistent dysbiosis [11] and maintaining high concentrations of circulating lipopolysaccharides (LPSs) over prolonged periods of time, which, in turn, promotes a chronic inflammatory environment, resulting in metabolic endotoxemia [12], which can lead to endotoxin tolerance that suppresses the expression of proinflammatory gene products [13] and impairs the normal process of host defense through a normal inflammatory response. 

In obese individuals, low-grade chronic inflammation may be exacerbated or synergistically exacerbated by an acute inflammatory response and inappropriate treatment for metabolic endotoxemia. Consequently, inflammation is of critical importance because of the multitude of chronic diseases associated with it. In this context, interest in the study of natural immunomodulators useful for the treatment of inflammation has recently in-creased [14]. These include functional foods, nutraceuticals, or bioactive compounds: compounds that, in addition to their nutritional contribution, help treat or prevent a dis-ease or disorder [15].

Garlic (Allium sativum) has been extensively studied for its antiviral and anti-inflammatory properties, which are largely attributed to its organosulfur compounds. These bioactive molecules have shown the ability to inhibit the replication of various viruses in preclinical studies [16]. In addition, the regular consumption of garlic and onion extract has been associated with a lower incidence of infectious respiratory diseases, particularly in the elderly [17], highlighting its potential as a supportive nutritional intervention to improve respiratory health.

Garlic (Allium sativum) proves to be a plant with extraordinary medicinal properties due to its bioactive compounds, especially organosulfur compounds such as allicin, diallyl disulphide, and ajoene. These substances have shown a wide range of therapeutic benefits, including antimicrobial, antioxidant, anticancer, antidiabetic, anti-inflammatory, and cardiovascular effects. Modern scientific studies have validated its use and confirmed its ability to prevent and treat chronic diseases such as cancer, cardiovascular disease, and diabetes. Garlic also helps to strengthen the immune system and regulate inflammation, making it a promising agent in the prevention of inflammatory and metabolic diseases such as obesity and metabolic syndrome. However, the importance of more thorough clinical research to determine safe and effective dosages must be emphasized [18,19].

Alliin, which is the S-allyl-cysteine-sulfoxide (Figure 1), is beginning to be considered a nutraceutical [20]. It is the main organosulfurized molecule of garlic that has been studied for its numerous beneficial physiological effects on health and disease [20]. Alliin exhibits anti-inflammatory properties both in vitro [21] and in vivo [20,22], and its effect on the microbiota in obesity models has also been described [23]. L-alliin can reduce adipocyte hypertrophy and, additionally, it can decrease the levels of major proinflammatory cytokines such as TNF-α, IL-6, and MCP-1 [21]. Consequently, this leads to an improvement in the immunometabolic profile, allowing for a reduction in blood glucose levels, representing an enhancement in the immunometabolic profile. In addition, it is interesting to note that the major water-soluble sulfur compound, S-allylcysteine (SAC), appears to have a direct inhibitory effect on IFN-γ and NF-κB and an indirect inhibitory effect on lipopolysaccharide (LPS)-induced IL-1β and TNF-α in human whole blood [24,25]. However, very little is known about the molecular mechanisms involved in the anti-inflammatory effect of garlic components [18]. 

On this basis, here, we have developed an experimental model to simulate two situations: health and chronic low-grade inflammation that occurs in obese individuals. Both are confronted with an acute inflammation-inducing challenge through the i.p. administration of LPS, mimicking an acute bacterial infection, to study the effect of L-alliin on the serum levels of cytokines when present in diet-induced obesity (DIO) or normally fed mice.

## 2. Materials and Methods

### 2.1. Diets 

A standard diet (STD) (4.07 kcal/g) with 18.3% protein, 22.1% fat, and 59.6% carbohydrate from TestDietTM (Richmond, VA, USA), diet 5755, was administered for the acclimatization phase and the standard groups. The high-fat diet (5.1 kcal/g) consisted of 18.1% protein, 61.6% fat, and 20.3% carbohydrate and was provided by TestDietTM (Richmond, VA, USA), diet 58Y1. The diets are indicated in the experimental design (Figure 2).

### 2.2. Alliin 

L-Alliin (S-allyl-L-cysteine sulfoxide), with the empirical formula C6H11NO3S, was synthesized by Sigma-Aldrich Quimica S de RL de CV (Hamburg, Germany), CAS number 17795-26-5, catalog number 74264. The alliin molecule can easily transform into other molecules during extraction from garlic, as alliinase can convert alliin into other derived by-products [26]. Nonetheless, the L-alliin molecule was purchased from Sigma Aldrich, certified to a purity of ≥90% by high-performance liquid chromatography (HPLC) and also stored lyophilized in amber glass to prevent the instability of the molecule. In addition, the L-alliin solution was prepared just before administration, which also prevents the instability of the molecules that could occur during the storage of the prepared solution.

### 2.3. Lipopolysaccharide

Escherichia coli lipopolysaccharide 0111: B4 was purchased from Millipore SigmaTM (Hamburg, Germany), catalog number: L3012, and was dissolved in PBS and administered intraperitoneally at a concentration of 0.1 mL per 10 g body weight according to the manufacturer’s instructions [27]. The average weight of the mice on STD and STD+A groups was 29 g, and for mice on HFD and HFD+A groups, it was 49 g.

### 2.4. Experimental Design

Previous studies using a C57BL/6J mouse model of diet-induced obesity through a high-fat diet have confirmed that the difference in body weight of mice from week 9 onwards is due to the accumulation of body fat, specifically in this strain of mice, and that this is also the case when reassessed at week 12 [28]. Furthermore, a weight gain of 16–23 g in C57BL/6J mice subjected to a 12-week HFD is thought to mimic human risk factors for the development of type 2 diabetes mellitus [15,29].

For this experiment, sixty 5-week-old male C57BL/6J mice were purchased from Morelos Vivarium (Morelos, Mexico). The protocol was approved by the Ethics Committee for Research and Biosafety of the Centro Universitario de Ciencias de la Salud, Universidad de Guadalajara (approval CI-02521). The animals were kept in the vivarium of the Centro Universtario de Ciencias de la Salud at a room temperature of approximately 26 °C and a 12 h light cycle and were provided with food and water ad libitum most of the time, except during the required fasting tests. After a 4-week acclimatization period and a body weight of approximately 25 g, the mice were randomly divided into 4 groups of 15 mice each using STATS 2.0TM Decision Analyst software. Two groups received a high-fat diet, and the other two groups received a standard diet for 12 weeks.

After 12 weeks, when obesity was reached in the high-fat diet groups, L-alliin was administered at a dose of 20 mg/kg mouse weight to one of these two groups and one of the two standard diet groups. The chosen dose of alliin for this model of acute inflammation was selected based on previous studies in which alliin lowered blood glucose levels at lower doses [20]. It was administered orally (using a straight stainless-steel needle for mice) at 15:30 Mexico City local time daily for four weeks with an isoosmolar 0.9% saline solution as the pharmacologic vehicle. The treatment groups received L-alliin, while the control groups received only the vehicle in the same amount (100 µL). 

Body weight and food intake were documented individually each week. Fasting glucose levels were determined at baseline (4 weeks), after the establishment of diet-induced obesity (12 weeks), after LPS administration, and on the same day before sacrifice at the end of the L-alliin treatment.

At the end of the fourth week of L-alliin treatment and 1 h before sacrifice, 5 mg/kg *E. coli* lipopolysaccharide (LPS) was administered intraperitoneally to all groups, and the mice were sacrificed by decapitation at the appropriate time points. The administration of LPS was chosen following the authors of [30], who have shown that a single dose of LPS can mimic the entry of bacterial products following intestinal barrier injury [30]. We chose this dose because it corresponds to half of the reported DL50.

### 2.5. Determination of Cytokines in Serum

For the evaluation of the treatment and control groups, 50 μL of serum was extracted from the whole blood of each mouse and stored in aliquots at −80 °C for the subsequent determination of serum cytokines. The Bio-Plex Pro Mouse Cytokine 23-Plex Panel (catalog no: M60-009RDPD) was used for the determination of IL-1α, IL-1β, IL-2, IL-3, IL-4, IL-5, IL-6, IL-9, IL-10, IL12p40, IL-12p70, IL-13, IL-17, eotaxin, G-CSF, GM-CSF, IFN-γ, Keratinocyte chemoattractant (KC), MCP-1, MIP-1α, MIP-1β, RANTES (regulated on activation, normal T-cell expressed and secreted), and TNF-α, according to the supplier’s instructions.

Essentially, Bio-Plex ProTM assays are immunoassays on magnetic beads. The assay is based on color-coded beads (microspheres), each of which is internally stained with a unique ratio of fluorescent dyes and coated with specific capture antibodies that correlate with the target analytes (cytokines and chemokines). The corresponding biomarker-specific capture antibodies are covalently bound to the beads. The coupled beads show a reaction with the biomarker-containing sample. To form a sandwich complex, a biotinylated detection antibody is added after a series of washes to remove unbound protein. The streptavidin–phycoerythrin conjugate is added to form the final detection complex. A fluorescent indicator or reporter is phycoerythrin. The Bioplex Manager software automatically calculates the concentrations using a non-linear mathematical model that is optimal for calculating different concentration ranges by interpolating the data with the concentration curves for each analyte. Data acquisition and analysis were performed using the Luminex MAGPIX system.

### 2.6. Statistical Analysis

The difference between groups was assessed using the one-way ANOVA test if the distribution was parametric; otherwise, the Kruskal–Wallis test was preferred. Differences were considered statistically significant at a *p*-value ≤ 0.05 and determined using the Tukey–Kramer multiple comparison test if they had a parametric distribution. Otherwise, the Dunn test was used and the statistical analyses were performed using GraphPad Prism software version 8.0.2 released on February 05, 2019.

## 3. Results

### Metabolic Effects of L-Alliin

The mice had a homogeneous weight at the beginning of the study, but after 7 weeks of different diets, both high-fat diet (HFD) groups had a significantly greater weight than the standard diet (STD) groups (*p* < 0.0001), which had a stable weight at the end of the study, without this being altered by treatment with alliin (Figure 3).

In previous studies, the administration of alliin alone did not reduce body weight in rats with DIO [20,23]. This effect was also observed in rats fed a standard diet. In contrast, Yu et al. [31] reported a statistically significant difference in body weight in rats receiving a standard diet with a higher dose of alliin (80 mg/kg body weight) in combination with quercetin (150 mg/kg body weight) or with quercetin alone. However, the observed effect on body weight could be due to possible interactions with components of the administered diet or the duration of administration and the dosage of the dietary supplement and not to a specific effect of allicin. It would probably be necessary to administer a higher dose of alliin or over a longer period to achieve an effect on body weight.

At the end of the L-Alliin treatment, the mean glucose values were not significantly different between the standard groups (STD+V and STD+A), while the values were significantly lower than those in the HFD groups (HFD+V and HFD+A) (*p* < 0.0001); there was also a statistically significant difference between the HFD+V and HFD+A groups (*p* < 0.0001) (Figure 4). 

The standard diet + L-Alliin group represents the effect of L-Alliin on a balanced diet. It is noteworthy that IL-1β and IL-2 were slightly increased in STD+V compared to STD+A (*p* < 0.002 and *p* < 0.02, respectively), while the opposite was observed for IL-6 (*p* < 0.02), IL-9 (*p* < 0.02), and G-CSF (*p* < 0.04) (Figure 5). 

Although the standard group is the reference for the homeostatic response to LPS, the difference in response between the HFD+V and HFD+A groups compared to the STD group is striking. When comparing the effects of diet type on the LPS response, increases in G-CSF (*p* < 0.04) and MIP-1β (*p* < 0.009) were observed in the HFD+V group. Meanwhile, compared with those in the HFD+V group, the levels of IL-2 (*p* < 0.02) (Figure 5a), IL-1β (*p* < 0.002), and TNF-α (*p* < 0.001) were significantly lower (Figure 5).

In addition, TNF-α was the only cytokine for which a difference was observed between the HFD groups (Figure 5g).

Finally, it is worth mentioning that there were cytokines whose changes were not statistically significant, but whose values showed interesting trends. The mean MCP1 and IFNγ levels were lower in the STD+V group than in the other groups (Figure 6a,b), while the eotaxin levels were comparable to those in the STD+A group (Figure 6c). A final observation is that IL-5 levels were slightly higher in the HFD+V group (Figure 6d). Results are summarized in Figure 7.

## 4. Discussion

The experimental design in this study aimed to simulate low-grade inflammation in diet-induced obesity followed by acute bacterial infection, and we expected impaired chemotaxis and dysregulated cytokine production because of the altered innate and adaptive response in obesity [32]. Treatment with L-Alliin resulted in a similar secretion of acute inflammatory mediators in the standard diet group and the two obese groups, whereas we had expected the opposite, namely, that L-Alliin would produce a comparable response to that of the standard diet control group, which has a healthy and physiological immune response. The activation of the inflammatory response is initially mediated by macrophages and neutrophils, which synthesize and secrete a variety of cytokines commonly described as acute inflammatory mediators [33]. However, in low-grade inflammation, the same cells and cytokines are involved, but there are no typical signs of inflammation or its consequences [34]. In this sense, organisms with this low-grade chronic inflammation may respond less well to any stimulus, whether exogenous or endogenous.

Nevertheless, the results obtained were not entirely unexpected. Previously, our research group had shown that L-alliin had a similar effect to LPS on TNF-α RNA expression in adipocytes, which behaved similarly to the control group in terms of secretion 12 h after the stimulus and only noticeably decreased after 24 h. In addition, Quintero et al. showed that IL-6 levels were higher in LPS-treated and alliin + LPS-treated adipocytes [21], which is consistent with our current results. In this case, L-alliin does not appear to have a specific effect on reducing the levels of proinflammatory cytokines, but rather on regulating other inflammatory processes.

The IL-2 and IL-1β levels observed in the STD+V group are particularly noteworthy. This control group had not received either the high-fat diet or the alliin treatment, suggesting that the observed results correspond to a physiologic response. Although the role of IL-2 and IL-1β in triggering the inflammatory response, cell recruitment and differentiation, and adaptive immunity in response to infectious stimuli has been extensively investigated in previous studies [35], fluctuations in cytokine levels have been associated with varying degrees of disease severity. In particular, some studies report that low levels of these cytokines correlate with increased disease severity and poor prognosis [36], while other studies suggest an inverse relationship [37].

On the other hand, the metabolic endotoxemia tolerance in the obese groups could be explained by their poor response to the LPS challenge. Therefore, the differential response to L-Alliin treatment in mice exposed to a high-fat diet could be due to the development of tolerance to endotoxemia. Prior exposure to LPS may attenuate the effects of an acute inflammatory response. This prior exposure may lead to the induction of specific miRNAs (miR-146, miR-125b, miR-98, miR-579, miR-132, and let-7e) due to gut dysbiosis, which subsequently exerts negative effects on the Toll-like receptor (TLR) signaling pathway [30].

Some inflammatory diseases could be alleviated by inducing the Th1 response while inhibiting the Th2 response [38]. Here, we have shown that alliin promotes serum Th1 cytokines, because the LPS challenge increased the levels of Th1 cytokines in the STD+A group, which facilitated an immune response. The initial increase in the levels of IFN-γ and TNF-α, which are the main regulators of the Th1 response, could indicate that the effect of L-Alliin in the obese group was similar to that in the standard group. While INF-γ induces the expression of chemoattractants such as MIG (gamma-inducible monokine), IP-10 (IFN-gamma-inducible protein 10), and I-TAC (IFN-inducible T cell-α chemoattractant), but also of MCP-1, MCP-2, and RANTES, and other cells such as macrophages, the main producers of TNF-α [39], this could mean that L-Alliin is able to help to trigger an appropriate inflammatory response to potential pathogens in an obese state.

Based on our findings, L-Alliin seems to somehow stimulate cellular chemotaxis by eliciting the release of key molecules, including IL-2, IFN-γ, TNF-α, MCP-1, IL-6, IL-9, and G-CSF (Figure 5 and Figure 6). This, in turn, mitigates the risk of severe inflammatory symptoms by preventing the release of IL-1β and its downstream molecules such as IL-1α, GM-CSF, and RANTES. Taken together, these results suggest that L-Alliin may act as an immunomodulator. However, the molecular mechanism involved remains unknown. 

In terms of molecular mechanisms, one possible signaling pathway involved in the action of alliin is the MAPK-NF-κB/AP-1/STAT-1 pathway [40]. Another interesting variant of action is the MAPK/ERK1/2 pathway. Quintero-Fabián et. al. [21] demonstrated that alliin treatment can suppress LPS-induced inflammatory signaling in vitro by generating an anti-inflammatory gene expression profile and altering the metabolic profile of adipocytes through reduced ERK1/2 phosphorylation. Molecular docking studies by Cheng et al. [41] later identified MAPK as a likely target for alliin binding. Furthermore, in vivo studies by Sánchez-Sánchez et al. [20] revealed that alliin treatment significantly reduced leptin and resistin levels in adipose tissue. Leptin, which interacts with glucagon-like peptide-1 (GLP-1) to regulate food intake and energy balance via different mechanisms [42,43], activates ERK1/2 in the arcuate nucleus of the hypothalamus [44], and GLP-1 also engages ERK signaling in diabetic mice [45]. These findings illustrate the intricate interplay between leptin, GLP-1, and ERK1/2 signaling in the regulation of energy homeostasis and suggest potential targets for the treatment of obesity through the combined activation of GLP-1 and leptin receptors in the central nervous system, all of which can be modulated by alliin.

Interestingly, DIO C57BL/6 mice infected with the influenza virus had a higher viral load and a lower survival rate than lean mice. Obese mice showed a weakened immune response characterized by a lower expression of IFN-a and IFNγ [46]. Moreover, TNFα expression was significantly higher in lean mice on the first day post-infection and reversed on day 5 post-infection. Similarly, our model showed an immediate response to LPS in both lean and DIO mice supplemented with L-alliin. It appears that L-alliin leads to a stronger immune response regardless of whether the mice were DIO or not. Namkoong also observed no response to IL-6 on the first day. Only on day 5 after infection did the DIO mice show a higher concentration than the lean mice. In agreement with the above, the administration of a methanolic garlic extract was able to significantly increase IFNγ serum levels in BALB/c mice infected with cystic echinococcosis [47]. Furthermore, cultured lymphocytes from BALB/c mice stimulated with garlic bulbs differentially regulate the expression of IFNγ [48].

Specifically, one of the most interesting findings were the increased levels of G-CSF and IL-9 in the STD+A group compared to those in the standard diet group. G-CSF is a cytokine that is secreted in response to LPS and TNF-α, and it is crucial for regulating neutrophil proliferation, differentiation, and survival, leading to a rapid increase in the number of circulating G-CSF-producing cells [49], suggesting that L-Alliin may act as an enhancer of the acute immune response in healthy individuals and show prophylactic effects against infections. While IL-9 is an important promoter of mast cell expansion, IL-1β, IL-6, and TGF-β production, and Treg cell production and plays an important role in allergies, parasitic infections, and autoimmunity, it has not been specifically described in obesity and related diseases; therefore, in this case, the immune response could also be triggered and enhanced in the presence of LPS.

In addition, this study confirmed that L-alliin can improve glucose levels in DIO mouse models [15,19]. The effect was associated with a reduction in β-cell apoptosis in a DM model, which was accompanied by the downregulation of nuclear factor kappa-B (NF-kB)/Toll-like receptor (TLR-4) and calcium ATPase/Ca of the sarco-endoplasmic reticulum [50]. In turn, NF-kB also regulates the proinflammatory cytokines IL-1, IL-2, IL-6, IL-8, IL-12, and TNFα [51]. Nonetheless, research has focused on the beneficial effects of garlic, not just alliin, on other metabolic diseases such as hypertension and blood lipid levels [52,53], and, more recently, garlic has even shown its potential as a dietary supplement for inflammatory responses [54]. This makes this dietary supplement attractive for further studies on this effect. In addition, systematic reviews have reported that garlic is a promising agent against metabolic syndrome and cardiovascular disease [55,56]. Therefore, further studies are needed to determine the molecules involved, their metabolism, safety, dosage, and indications as a drug.

## 5. Conclusions

An appropriate immune response is crucial for a physiological inflammatory process. Therefore, the experimental design of this study aimed to simulate first a state of low-grade inflammation due to diet-induced obesity and then an acute bacterial infection (mimicked by LPS administration). Consequently, we have shown here that this first response, when induced by LPS, was promoted by L-alliin in the standard diet group and that it was also induced in the obese group, albeit to a lesser extent, probably due to some tolerance to endotoxemia. We observed that pretreatment with alliin increased the concentration of acute inflammatory cytokines in response to LPS stimulation in normal-weight animals, whereas this increase was lower in HFD animals and was due to a few key cytokines. This suggests that alliin can enhance the initial response to acute inflammatory stimuli in normal-weight animals and that this effect is also present in DIO animals, albeit to a lesser extent. However, future research on these prophylactic effects on acute inflammation are needed. In addition, from the perspective of this study, we could consider analyzing experimental groups without the LPS challenge, which showed only the effects of the interventions in relation to low-grade chronic inflammation. Similarly, the determination of anti-inflammatory cytokines may shed light on the counterregulatory mechanisms of inflammation. Another consideration that must not be disregarded is the dose of L-Alliin used, although other studies have used different concentrations with similar results [20,23,24].

Finally, our findings demonstrate that L-alliin can enhance immunity in healthy organisms and function as an immunomodulator in low-grade inflammation. Additionally, L-Alliin has exhibited beneficial effects on metabolism, such as improving glucose levels in diet-induced obesity. Future research on plant-derived immunomodulators should focus on deciphering the cellular and molecular mechanisms underlying the immunomodulatory effects of natural and biologically active metabolites, identifying novel promising candidates for future immunotherapeutic strategies.

## Figures and Tables

**Figure 1 metabolites-14-00580-f001:**
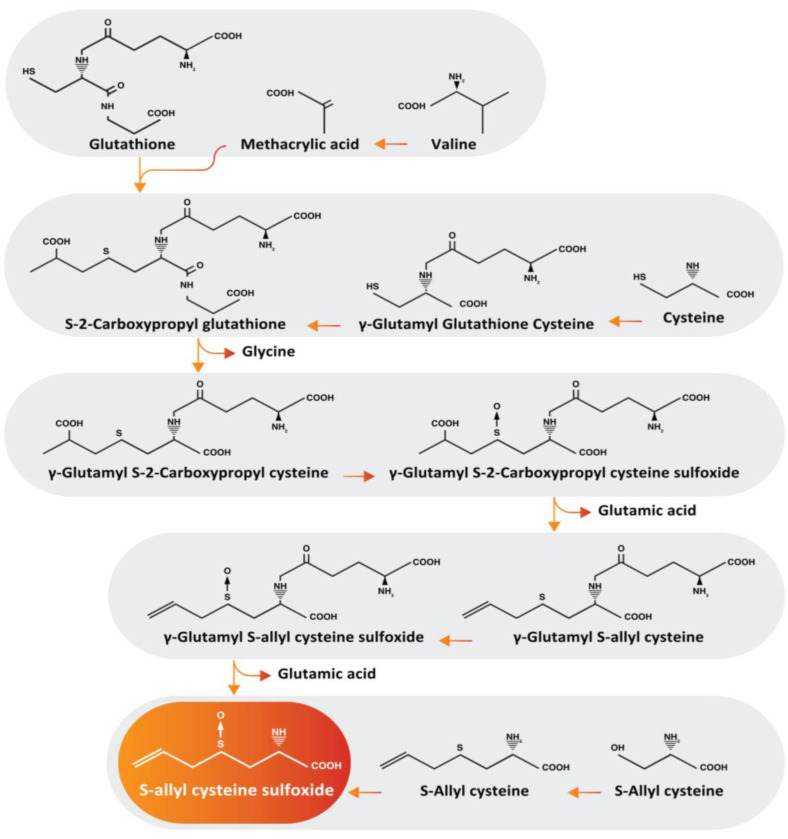
The biosynthesis of alliin. There are various metabolic pathways for the biosynthesis of the alliin molecule. The precursors of alliin include glutathione, valine, cysteine, and serine. S-2-carboxypropylglutathione is produced from glutathione, a tripeptide consisting of glycine, cysteine, and glutamic acid, and methacrylic acid derived from valine. Since garlic contains γ-glutamylcysteine, cysteine can also be used to form S2-carboxypropylglutathione, which is converted to γ-glutamyl-S-2-carboxypropylcysteine through the cleavage of glycine. The flavin-containing monooxygenase in the cytosol is responsible for S-oxidation in the biosynthesis of γ-glutamyl-S-2-carboxypropylcysteine sulfoxide in the presence of nicotinamide adenine dinucleotide phosphate and flavin adenine dinucleotide. Since the carboxypropyl group is formed by decarboxylation and oxidation, γ-glutamyl-S-allylcysteine can be obtained from γ-glutamyl-S-2-carboxypropylcysteine sulfoxide with the release of glutamic acid as a by-product. When γ-glutamyl-S-allylcysteine is S-oxidized, it forms γ-glutamyl-S-allylcysteine sulfoxide, but when S-allylcysteine is S-oxidized, it forms allicin.

**Figure 2 metabolites-14-00580-f002:**
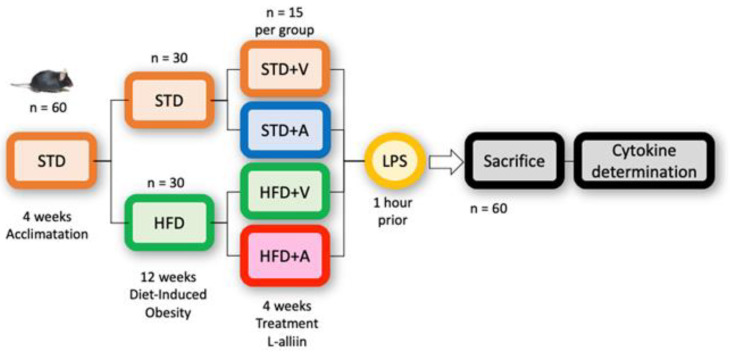
Experimental design.

**Figure 3 metabolites-14-00580-f003:**
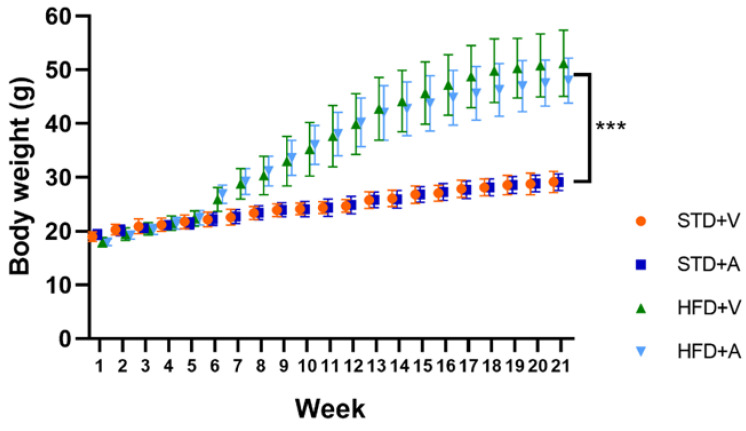
Body weight of male C57BL/6J mice fed a standard diet (STD) or a high-fat diet (HFD) and alliin (A) or vehicle (V) (*n* = 15 in each group), recorded over 21 weeks. Mean and standard deviation (SD) are shown. *** *p* < 0.001.

**Figure 4 metabolites-14-00580-f004:**
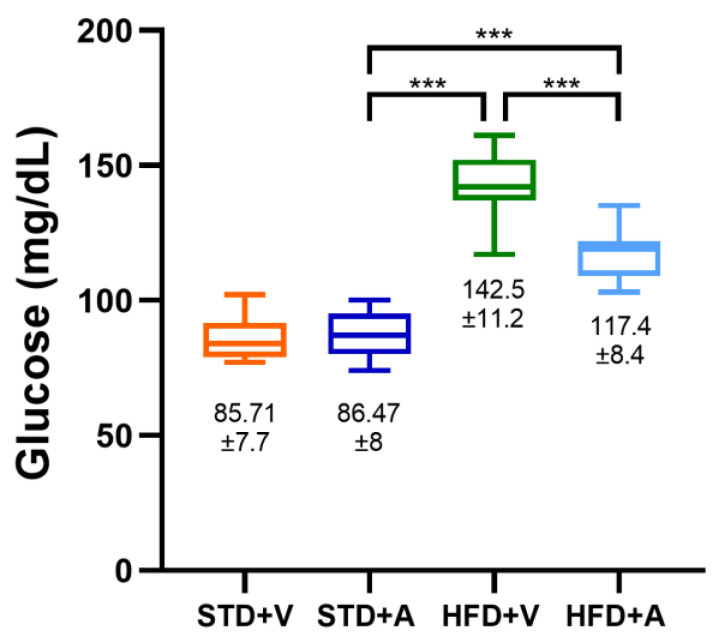
Glucose serum levels at the end of the L-Alliin treatment (at 20 weeks of age) of male C57BL/6J mice fed a standard diet (STD) or a high-fat diet (HFD) and alliin (A) or vehicle (V) (n = 15 in each group), recorded at 21 weeks of age. Mean values and standard deviations (SD) are shown. *** *p* < 0.001.

**Figure 5 metabolites-14-00580-f005:**
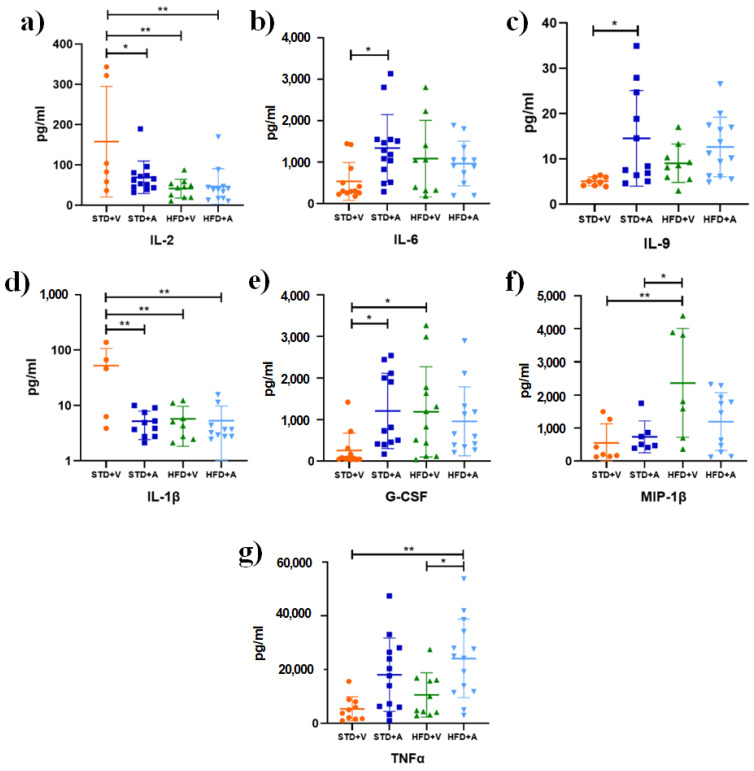
Effect of an obesogenic diet and treatment with L-Alliin on cytokine levels in the acute phase. (**a**) IL-2 (**b**) IL-6, (**c**) IL-9, (**d**) IL-1β, (**e**) G-CSF, (**f**) MIP-1β, and (**g**) TNF-α. Male C57BL/6J mice fed a standard diet (STD) or a high-fat diet (HFD) and supplemented with alliin (A) or vehicle (V). Mean and standard deviation (SD) are shown. * *p* < 0.05, ** *p* < 0.01.

**Figure 6 metabolites-14-00580-f006:**
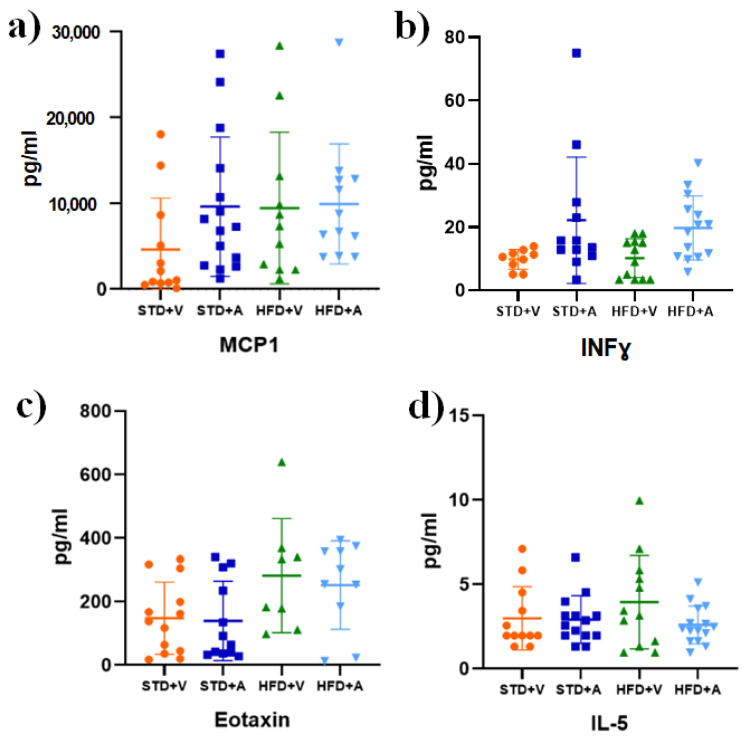
Cytokines that were not statistically significantly influenced by diet or L-alliin. (**a**) MCP-1, (**b**) INF, (**c**) Eotaxin, and (**d**) IL-5. Male C57BL/6J mice fed a standard diet (STD) or a high-fat diet (HFD) and supplemented with alliin (A) or vehicle (V). Mean and standard deviation (SD) are shown.

**Figure 7 metabolites-14-00580-f007:**
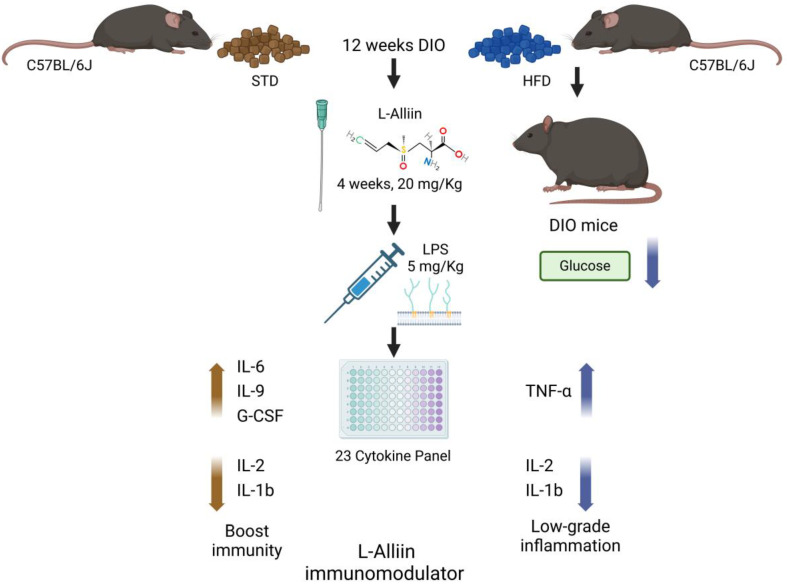
L-Alliin as immunomodulator in both standard diet (STD) and high-fat diet (HFD) obesity-induced (DIO) mice. After 12 weeks of diet, two groups received L-alliin (20 mg/Kg) via a gastric cannula for four weeks. Later, an acute lipopolysaccharide (LPS) stimulus (5 mg/Kg) was administered, and then the glucose levels and serum levels of 23 cytokines were determined. L-Alliin acts as an immunomodulatory molecule by stimulating the release of key molecules in both normal weight and obesity and somehow preventing the release of molecules involved in severe inflammatory response.

## Data Availability

The raw data supporting the conclusions of this article will be made available by the authors on request.

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
