# Peer review of "Modulatory L-Alliin Effect on Acute Inflammatory Cytokines in Diet-Induced Obesity Mice"

_metabolites, 2024, doi:10.3390/metabo14110580_

Round 1
Reviewer 1 Report
Comments and Suggestions for Authors
Dear authors,
Please see the documents as attached.

The quality of the English language is good and minor editing are required.
Author Response
Reviewer #1 Comments
General comments:
The manuscript is well written. The authors investigated the “Modulatory L-Alliin Effect on
Acute Inflammatory Cytokines in DIO Mice”. The study is interesting and adds to the existing body of knowledge. The data and results of the current study are well presented and the findings are beneficial to the pharmaceutical industry. The current study has shown that L-Alliin stimulates cellular chemotaxis by eliciting the release of key molecules including IL-2, IFN-γ, TNF-α, MCP-1, IL-6, IL-9, and G-CSF. However, there are a few things that need clarification and revisions.
ANSWER: We would like to thank the reviewer for his positive comments on our work and for the time taken to review our manuscript in such detail. Below you will find the answers to the reviewer's queries.
Details comments:
- Page 1, Lines 28-29: Abstract; Please include the objective of the current study. Write the full name for DIO mice then abbreviate it
ANSWER: full name for DIO mice has been included and also the objective of the study as follows: Objective: to improve scientific evidence for the immunomodulatory role of alliin in obesity and inflammation.
- Page 2, Introduction: Please add the latest prevalence of non-communicable diseases, particularly obesity and cardiovascular diseases. Please add more citations regarding the use of garlic particularly its current pharmacological effects
ANSWER: We thank the reviewer for the opportunity to go deep into this aspect. In agreement with the request, we have included a whole new first paragraph about the prevalence of obesity and cardiovascular diseases as follows:
“Since 1990, global adult obesity has more than doubled, with 43.5% of individuals aged 18 and over classified as overweight [1] and 16% as obese [2]. In the same fashion, since 1990, the worldwide prevalence of deaths attributed to cardiovascular diseases has increased by 61% (from 12,345,727 to 19,906,615) [3]. Therefore, these diseases and their complications have become an enormous burden on health services and a worldwide public health problem.”
In addition, we have included a new paragraph on the use of garlic, particularly its pharmacological effects as follows:
Garlic (Allium sativum) proves to be a plant with extraordinary medicinal properties due to its bioactive compounds, especially organosulphur compounds such as allicin, diallyl disulphide and ajoene. These substances have shown a wide range of therapeutic benefits, including antimicrobial, antioxidant, anticancer, antidiabetic, anti-inflammatory and car-diovascular effects. Modern scientific studies have validated its use and confirmed its ability to prevent and treat chronic diseases such as cancer, cardiovascular disease and diabetes. Garlic also helps to strengthen the immune system and regulate inflammation, making it a promising agent in the prevention of inflammatory and metabolic diseases such as obesity and metabolic syndrome. However, the importance of more thorough clinical research to determine safe and effective dosages must be emphasized
- Page 4, Materials and methods: The heading subtitle in this section should be revised to 2.1. Diet; 2.2. Alliin; 2.3. Lipopolysaccharide; 2.4. Experimental design; 2.5. Determination of cytokines in serum; 2.6. Statistical analysis.
ANSWER: Numbering of headlines have been revised and corrected accordingly.
- Page 4, Line 141: A standard diet (STD).
ANSWER: Done
- Page 5, Lines 156-157: Please mention the weight of mice, total of mice used, and age of mice (week).
ANSWER: We thank the reviewer for pointing out these relevant aspects. We have added the data requested as follows:
New lines 171-172 “The average weight of the mice on STD and STD+A groups was 29 g and for mice on a HFD and HFD+A groups it was 49 g.”
New line 180 “For this experiment, sixty male C57BL/6J mice of 5-week-old”
- Page 5, Lines 189-191: Statistical analysis; Please mention the data presentation of the current study. e.g. Data was presented as mean ± standard deviation (SD).
ANSWER: As the reviewer correctly mentioned, the data is presented as mean and standard deviation where appropriate. This clarification has been added on new lines 231-236
- Please add the conclusion/summary of the current study.
ANSWER: We thank the reviewer for the opportunity to address this aspect in a separate section. Consequently, we have added section 5 of the manuscript as follows:
- Conclusion
An appropriate immune response is crucial for a physiological inflammatory process. Therefore, the experimental design of this study aimed to simulate first a state of low-grade inflammation due to diet-induced obesity and then an acute bacterial infection (mimicked by LPS administration). Consequently, we have shown here that this first response, when induced by LPS, was promoted by L-alliin in the standard diet group and that it was also induced in the obese group, albeit to a lesser extent, probably due to some tolerance to endotoxemia. We observed that pretreatment with alliin increased the concentration of acute inflammatory cytokines in response to LPS stimulation in normal weight animals, whereas this increase was lower in HFD animals and was due to a few key cytokines. This suggests that alliin can enhance the initial response to acute inflammatory stimuli in normal weight animals and that this effect is also present in DIO animals, albeit to a lesser extent. However, further studies on these prophylactic effects on acute inflammation are needed. Finally, we have shown that L-alliin can boost immunity in healthy organisms and act as an immunomodulator in low-grade inflammation. In addition, L-alliin has been shown to have positive effects on metabolism, e.g. on glucose levels, in diet-induced obesity.

Reviewer 2 Report
Comments and Suggestions for Authors
The manuscript presents an interesting study on the modulatory effects of L-Alliin on acute inflammatory cytokines in diet-induced obesity (DIO) mice. The subject matter is relevant and contributes to the growing field of natural compounds and their therapeutic potential in modulating inflammation, particularly in obesity-related conditions. However, there are several areas that need improvement to enhance the clarity and scientific rigor of the article.
Some minor issues have to be fixed. Indeed, the methodology section requires significant expansion. The current description of experimental protocols lacks detail, making it difficult for readers to assess the reproducibility of the study. It would be helpful to include more specific information regarding the preparation and administration of L-Alliin, the dosage used, and the criteria for the selection of mice in the DIO model. Furthermore, the description of the cytokine measurements should be more thorough, including the assay techniques, controls used, and statistical analysis methods employed to ensure the validity of the results.
The results section provides some valuable insights, but further discussion and interpretation of the findings are necessary. The current presentation of the data is somewhat limited, and a deeper analysis of the significance of the observed changes in cytokine levels is warranted. It would strengthen the manuscript to explore the potential mechanisms through which L-Alliin exerts its modulatory effects, as well as any broader implications these findings might have for inflammation and obesity treatment. Additionally, including more graphical representations of the data could improve the reader's understanding of the results.
The manuscript currently lacks a clear conclusion section. It is important to summarize the key findings and discuss their relevance within the context of existing literature. The conclusion should also address any limitations of the study and suggest possible directions for future research. This will help to consolidate the significance of the work and provide a clearer takeaway message for the readers.
Comments on the Quality of English LanguageThe overall quality of the English in the manuscript is quite good, with only minor issues that do not affect readability. I recommend a general, light revision to improve the flow and clarity in a few places, but no major language corrections are necessary. This will ensure that the manuscript is polished and professionally presented.
Author Response
Reviewer #2 Comments
The manuscript presents an interesting study on the modulatory effects of L-Alliin on acute inflammatory cytokines in diet-induced obesity (DIO) mice. The subject matter is relevant and contributes to the growing field of natural compounds and their therapeutic potential in modulating inflammation, particularly in obesity-related conditions. However, there are several areas that need improvement to enhance the clarity and scientific rigor of the article.
ANSWER: We would like to thank the reviewer for his positive comments on our work and for the time taken to review our manuscript. Below you will find the ANSWERs to the reviewer's suggestions.
- Some minor issues have to be fixed. Indeed, the methodology section requires significant expansion. The current description of experimental protocols lacks detail, making it difficult for readers to assess the reproducibility of the study. It would be helpful to include more specific information regarding the preparation and administration of L-Alliin, the dosage used, and the criteria for the selection of mice in the DIO model.
ANSWER: We thank the reviewer for the opportunity to elaborate on this methodological aspect. We have included the following more detailed information in response to this aspect:
In lines 190-197: “After 12 weeks, when obesity was reached in the high-fat diet groups, L-alliin was administered at a dose of 20 mg/kg mouse weight to one of these two groups and one of the two standard diet groups. The chosen dose of alliin for this model of acute inflammation was selected based on previous studies in which alliin lowered blood glucose levels at lower doses (Sánchez-Sánchez et al., 2020]. It was administered orally (using a straight stainless-steel needle for mice) at 15:30 local time Mexico City, daily for four weeks with an isoosmolar 0.9% saline solution as the pharmacologic vehicle. The treatment groups received L-alliin, while the control groups received only the vehicle in the same amount (100 µL).
Criteria for the selection of mice for the DIO model.
In lines 174-179: Previous studies using C57BL/6J mice model of diet-induced obesity through a high-fat diet have confirmed that the difference in body weight of mice from week 9 onwards is due to the accumulation of body fat, specifically in this strain of mice, and that this is also the case when reassessed at week 12 (Odom et al, 2022). Furthermore, a weight gain of 16-23 g in C57BL/6J mice subjected to a 12-week HFD is thought to mimic human risk factors for the development of type 2 diabetes mellitus (Sánchez-Sánchez et al., 2020; Nguyen-Phuong et al., 2023).”
- Furthermore, the description of the cytokine measurements should be more thorough, including the assay techniques, controls used, and statistical analysis methods employed to ensure the validity of the results.
ANSWER: We thank the reviewer for the opportunity to go deep into the information on techniques and statistical analysis.
In lines 209-229: “For the evaluation of the treatment and control groups, 50 μl of serum was extracted from the whole blood of each mouse and stored in aliquots at −80°C for subsequent determination of serum cytokines. The Bio-Plex Pro Mouse Cytokine 23-Plex Panel (catalog no: M60-009RDPD) was used for the determination of IL-1α, IL-1β, IL-2, IL-3, IL-4, IL-5, IL-6, IL-9, IL-10, IL12p40, IL-12p70, IL-13, IL-17, eotaxin, G-CSF, GM-CSF, IFN-γ, Keratinocyte chemoattractant (KC), MCP-1, MIP-1α, MIP-1β, RANTES (regulated on activation, normal T-cell expressed and secreted) and TNF-α, according to the supplier's instructions.
Basically, Bio-Plex ProTM assays are immunoassays on magnetic beads. The assay is based on color-coded beads (microspheres), each of which is internally stained with a unique ratio of fluorescent dyes and coated with specific capture antibodies that correlate with the target analytes (cytokines and chemokines). The corresponding biomarker-specific capture antibodies are covalently bound to the beads. The coupled beads show a reaction with the biomarker-containing sample. To form a sandwich complex, a biotinylated detection antibody is added after a series of washes to remove unbound protein. The streptavidin-phycoerythrin conjugate is added to form the final detection complex. A fluorescent indicator or reporter is phycoerythrin. The Bioplex Manager software automatically calculates the concentrations using a non-linear mathematical model that is optimal for calculating different concentration ranges by interpolating the data with the concentration curves for each analyte. Data acquisition and analysis were performed using the Luminex MAGPIX system.
In lines 231-236: “The difference between groups was assessed using the one-way ANOVA test if the distribution was parametric, otherwise the Kruskal–Wallis test was preferred. Differences were considered statistically significant at a P-value ≤ 0.05 and determined using the Tukey-Kramer multiple comparison test if they had a parametric distribution. Otherwise, the Dunn test was used and the statistical analyzes were performed using GraphPad Prism 8.0.2. software.”
- The results section provides some valuable insights, but further discussion and interpretation of the findings are necessary. The current presentation of the data is somewhat limited, and a deeper analysis of the significance of the observed changes in cytokine levels is warranted. It would strengthen the manuscript to explore the potential mechanisms through which L-Alliin exerts its modulatory effects, as well as any broader implications these findings might have for inflammation and obesity treatment.
ANSWER:
In terms of molecular mechanisms, one possible signaling pathway involved in the action of alliin is the MAPK/ERK1/2 pathway. Quintero-Fabián et al (2013) demonstrated that alliin treatment can suppress LPS-induced inflammatory signaling in vitro by generating an anti-inflammatory gene expression profile and altering the metabolic profile of adipocytes through reduced ERK1/2 phosphorylation. Molecular docking studies by Cheng et al. (2022) later identified MAPK as a likely target for alliin binding. Furthermore, in vivo studies by Sánchez-Sánchez et al. (2020) revealed that alliin treatment significantly reduced leptin and resistin levels in adipose tissue. Leptin, which interacts with glucagon-like peptide-1 (GLP-1) to regulate food intake and energy balance via different mechanisms (Zhao et al., 2011; Poleni et al., 2012), activates ERK1/2 in the arcuate nucleus of the hypothalamus (Rahmouni et al., 2009), and GLP-1 also engages ERK signaling in diabetic mice (Jolivalt et al., 2012). These findings illustrate the intricate interplay between leptin, GLP-1 and ERK1/2 signaling in the regulation of energy homeostasis and suggest potential targets for the treatment of obesity through the combined activation of GLP-1 and leptin receptors in the central nervous system, all of which can be modulated by alliin.
- Additionally, including more graphical representations of the data could improve the reader's understanding of the results.
ANSWER: We thank the reviewer for this suggestion. Accordingly, we can include a summarizing figure (graphical summary) of the results obtained for better illustration and understanding.
- The manuscript currently lacks a clear conclusion section. It is important to summarize the key findings and discuss their relevance within the context of existing literature. The conclusion should also address any limitations of the study and suggest possible directions for future research. This will help to consolidate the significance of the work and provide a clearer takeaway message for the readers.
ANSWER: We thank the reviewer for the opportunity to comment on this item. In line with this comment, we have added a concluding section 5 with limitations, conclusions, and perspectives.
